**MS No.:** bg-2018-30
**Title:** Basin-scale variability of microbial methanol uptake in the Atlantic Ocean
**Author(s):** Stephanie L. Sargeant et al.
**MS Type:** Research article
Dear Silvio Pantoja,
Please see below for a point by point reply to all the reviewer's comments and your own minor
corrections. We also include a revised manuscript showing all the track changes. Please note that the
references have been amended to comply with Biogeosciences format in the finalised version only.
Yours sincerely,
Joanna Dixon, Stephanie Sargeant & Colin Murrell
**Response to Reviewers Comments** (Authors responses are in bold)
**L. Chistoserdova (milachis@uw.edu)**
This study provides an inventory of measurements relevant to methanol consumption by
microbial communities across the Atlantic, a rare basin-wide evaluation. The description is
somewhat monotonous, but it is what it is. A great variability is uncovered across provinces
and across depths, but little correlation is found of methanol oxidation/assimilation with
respect to where it happens. In general, dissimilation is somewhat correlated with the
presence of SAR11, and, in general, assimilation is two orders of magnitude lower than
dissimilation. Which means SAR11 uses some other carbon source(s) for building biomass,
and these remain unknown. In general, I think, even if many questions remain unanswered,
this is a useful benchmark study. – **We thank L. Chistoserdova for their comments.**
Improvements that I would like to suggest: – **Each of L. Chistoserdova's comments has**
**been addressed individually as follows.**
Page 14, line 15, please say tetrahydrofolate-linked C1 transfer pathway, there are various
oxidation levels and none of them are methyl- level after methanol oxidation. - **"methyl-THF**
**linked oxidation pathway" will be changed to "tetrahydrofolate-linked C1 transfer**
**pathway" (Page 14, line 15).**
Same page, lines 15-18. You do not see any bona fide methylotrophs in your 16S libraries.
How can you conclude that they are active, along with SAR11? Either elaborate or remove
this statement. PCR amplification of specific genes does not compare with 16S analysis, and
you do not do any in this study anyway. – **Specific gene amplification, using *mxaF***
**functional gene primers, has been conducted previously on the same samples as the**
**current study looking at 16S rRNA analysis. The *mxaF* functional gene analysis**
**identified classic methylotrophic bacteria from these samples, these results are**
**published in a previous manuscript Dixon et al. (2013). To clarify this in the text we**
**have amended "Methylotrophic bacteria such as *Methylophaga* sp., *Methylococcaceae***
***sp.* and *Hyphomicrobium* sp. have been previously identified, using *mxaF* functional**
**gene primers (which encode for the classical methanol dehydrogenase), from the upper**
**water column of Atlantic Ocean provinces (Dixon et al., 2013)" to "Previously**
**methylotrophic bacteria such as *Methylophaga* sp., *Methylococcaceae* sp. and**
***Hyphomicrobium* sp. have been identified, using *mxaF* functional gene primers (which**
**encode for the classical methanol dehydrogenase), from the same DNA samples**
**analysed for 16S rRNA genes in this study, from the upper water column of Atlantic**
**Ocean provinces (Dixon et al., 2013)" (Page 14, lines 7-10).**
Meantime, an interesting question: while true methylotrophs do inhabit marine waters, why
are they so sparse and apparently uncompetitive compared to SAR11? Can you elaborate? –
**The authors agree this is an interesting question and more work is needed to unpick**
**this. We don't have an answer for this with the published literature and knowledge**
**currently available, however we can speculate that it may be down to the shear**
**abundance and evolutionary strategy of SAR11 in comparison to true methylotrophs.**
**SAR11 are the most abundant, free living, heterotrophic bacteria in open ocean systems**
**and are often the most abundant organisms in oligotrophic waters. The competitiveness**
**and high abundance of SAR11 cells in open ocean waters could be one part of a reason**
**why true methylotrophs are relatively sparse in comparison. SAR11 have been shown**
**to have one of the smallest genome sizes of any replicating cell and Giovannoni et al.,**
**(2005) suggest that the streamlining hypothesis may provide an explanation for this.**
**The streamline hypothesis, assumption that selection reduces genome size due to the**
**metabolic burden of replicating DNA without adaptive value, could be the strategy**
**responsible for the dominance and success of the SAR11 clade in oligotrophic waters.**
(this has not been added to the manuscript)
Fig. 5 would greatly benefit from introducing colors, would be so much easier to compare
guild distribution. Also, please order the taxa in a uniform way, i.e. use the same taxon order
in each panel. **– We have made the appropriate changes as suggested.**
Table 1. Specify that you show ranges below averages/means, specify which. Specify what
NA means. **– A comment has been added to the end of the Table caption to clarify what**
**these values are "Values given are average ± standard deviation (range). NA denotes**
**that data is not available."**
**S. Giovannoni (Referee) steve.giovannoni@oregonstate.edu**

This is a valuable study that adds significantly to our understanding of methanol oxidation in the oceans. The authors report seawater methanol oxidation rates obtained with 14C tracer methods, and microbial diversity measurements, from a latitudinal transect between 40S and 50N. They find that methanol oxidation rates are correlated with SAR11 relative abundance. Overall, the reported rates of methanol oxidation are in good agreement with previous measurements, but this study is exceptional in geographical scope and exploration of variables such as community composition and depth. Interestingly, the manuscript reports an inverse correlation between bacterial production estimated by the 3H leucine method and methanol oxidation. Although there have been a number of reports previously on methanol cycling in the oceans, I see the subject coming of age with this report, which confirms what we knew and also shows us new trends that could only have been observed by making extensive measurements across a latitudinal transect. - **We thank S. Giovannoni for his comments.**

A couple of comments follow about aspects of the paper that could be improved. - **Each of S. Giovannoni's comments has been addressed individually as follows.**

1. I recommend commenting on the abundance of methylophaga and OM43 in the 454 data, or indicate they were not detected if that is the case. It may be that the relatively low coverage obtained in this study (386 seqs/sample) led to these taxa being undetectable. If this is the case, that should be explained so that readers new to this topic understand the issues. OM43 is not mentioned at all, but perhaps it should be, since it has been shown to be an obligate methylotroph, is one of the dominant taxa in some coastal environments, and has been shown to be a source of abundant XoxF peptides in a coastal ocean metaproteome. – **The authors recognise this omission in the manuscript and acknowledge that these taxa**

**should be included to reflect our current understanding of marine methanol utilisation.**
**Therefore, we have added the following;**
**"Although numerically very rare (1-11 16S rRNA gene sequences per sample), 16S**
**rRNA gene sequences identified as *Methylophaga* spp., *Methylophaga* sp. DMS021**
**(EU001861) and uncultured *Methylophaga* sp. (EU031899), were found in each of the**
**Atlantic Ocean provinces in this study (at 97% PAR or 200m depth), consistent with**
**previous identification of *Methlophaga* spp. in these Atlantic provinces using *mxaF* gene**
**cloning in (Dixon et al., 2013)." (Page 14, lines 14-19).**
**"Members of *Betaproteobacteria*, OM43, have been shown to be obligate methylotrophs,**
**with cultivated cells of strain HTCC2181 dissimilating 3.5 times more methanol than**
**was assimilated (Halsey et al., 2012). OM43 were not successfully identified in the 16S**
**rRNA sequences in this study, which could be an artefact of the relatively low sequence**
**coverage (386 sequences per sample) leading to this taxon not being detectable. During**
**a previous coastal study in the western English Channel (16S rRNA pyrosequence data,**
**Sargeant et al., 2016) only a single sequence of the OM43 clade, HTCC2181, was**
**identified. This is a limitation of this type of environmental sequencing effort and**
**should be a consideration in planning any future projects aiming to understand**
**microbial function through process measurements alongside the generation of**
**metagenomic datasets." (Page 14, line 28 – Page 15, line 6).**
**We have also added the additional reference: Halsey KH, Carter AE, Giovannoni SJ**
**(2012) Synergistic metabolism of a broad range of C1 compounds in the marine**
**methylotrophic bacterium HTCC2181. Environmental Microbiology 14:630-640.**
2. Amplicon ratios are not as powerful as cell numbers for identifying correlations between
taxa and rates, although they are much easier to obtain. So, the correlations with SAR11 are
not with SAR11 cells per unit volume, which would be best, but rather a correlation between
the relative success of SAR11 in the community and rates of MeOH oxidation. I suggest the
authors revisit the manuscript and choose wording that conveys these issues to oceanographer
readers, who often misunderstand this aspect of relative abundance data. **– The authors**
**recognise that this is a limitation and clarity should be provided.  We have added;**
**"It should be noted that this correlation has been made with amplicon ratios, relating to**
**the relative success of SAR11 in the community, rather than with SAR11 cell numbers**
**specifically." (Page 13, lines 10-12).**
**"More work is required to add clarity and understanding to the role that SAR11 cells**
**play in marine community methanol dissimilation." (Page 14, lines 3-4).**
**Associate Editor (minor comments)**
Page 10, line 20. Replace "calculated" with "found" or "detected", or something like that.
**Done**.
Legend Fig. 1 replace "circles" with "ovals" **Done.**
**We hope the manuscript is now acceptable,**
**Best wishes**
**Jo**
**Dr Joanna L Dixon**
# Basin-scale variability of microbial methanol uptake in the
# Atlantic Ocean
Stephanie L. Sargeant[1*], J. Colin Murrell[2], Philip D. Nightingale[1], Joanna L. Dixon[1]
[1]Plymouth Marine Laboratory, Prospect Place, West Hoe, Plymouth, Devon, PL1 3DH, UK
[2]School of Environmental Science, University of East Anglia, Norwich Research Park, Norwich, NR4 7TJ, UK
*Now at: Centre for Research in Biosciences, University of the West of England, Frenchay Campus,
Coldharbour Lane, Bristol, BS16 1QY, UK.
*Correspondence to*: Joanna L. Dixon (jod@pml.ac.uk)
**Abstract.** Methanol is a climate active gas and the most abundant oxygenated volatile organic compound
(OVOC) in the atmosphere and seawater. Marine methylotrophs are aerobic bacteria that utilise methanol from
seawater as a source of carbon (assimilation) and/or energy (dissimilation). A few spatially limited studies have
previously reported methanol oxidation rates in seawater; however the basin-wide ubiquity of marine microbial
methanol utilisation remains unknown. This study uniquely combines seawater $^{14}$C labelled methanol tracer
studies with 16S rRNA pyrosequencing to investigate variability in microbial methanol dissimilation and known
methanol utilising bacteria throughout a meridional transect of the Atlantic Ocean between 47$^o$ N to 39$^o$ S.
Microbial methanol dissimilation varied between 0.05–1.68 nmol $l^{-1}$ $h^{-1}$ in the top 200 m of the Atlantic Ocean
and showed significant variability between biogeochemical provinces. The highest rates of methanol
dissimilation were found in the northern subtropical gyre (average 0.99±0.41 nmol $l^{-1}$ $h^{-1}$), which were up to
eight times greater than other Atlantic regions. Microbial methanol dissimilation rates displayed a significant
inverse correlation with heterotrophic bacterial production (determined using $^3$H-leucine). Despite significant
depth stratification of bacterial communities, methanol dissimilation rates showed much greater variability
between oceanic provinces compared to depth. There were no significant differences in rates between samples
collected under light and dark environmental conditions. The variability in the numbers of SAR11 (16S rRNA
gene sequences) were estimated to explain approximately 50% of the changes in microbial methanol
dissimilation rates. We estimate that SAR11 cells in the Atlantic Ocean account for between 0.3-59 % of the
rates of methanol dissimilation in Atlantic waters, compared to <0.01-2.3 % for temperate coastal waters. These
results make a substantial contribution to our current knowledge and understanding of the utilisation of
methanol by marine microbial communities, but highlight the lack of understanding of *in situ* methanol
production mechanisms.
**1. Introduction**
Methanol is the most abundant oxygenated volatile organic compound (OVOC) in the
background troposphere where it acts as a climate active gas, influencing the oxidative
capacity of the atmosphere, concentrations of ozone and hydroxyl radicals (Carpenter et al.,
2012). Methanol has been shown to be ubiquitous in waters of the Atlantic Ocean ranging

between <27–361 nM (Beale et al., 2013; Williams et al., 2004; Yang et al., 2013; Yang et al., 2014). Our knowledge of the sources and sinks of methanol is limited and often lacks consensus. For example, recent eddy covariance flux estimates demonstrated a consistent flux of atmospheric methanol into the surface waters of a meridional transect of the Atlantic Ocean (Yang et al., 2013). However, along a similar transect, 12 months earlier, Beale et al. (2013) calculated that the Atlantic Ocean represents an overall source of methanol to the atmosphere (3 Tg $yr^{-1}$), which was largely attributable to an efflux from the North Atlantic gyre; where surface concentrations were as high as 361 nM. Wet deposition from rainwater has also recently been suggested to represent a supply of methanol to the ocean (Felix et al., 2014).

Although *in situ* marine photochemical production of methanol has previously been found to be insignificant (Dixon et al., 2013), there is thought to be a substantial unidentified biological source of methanol in seawater (Dixon et al., 2011a). Biological production by phytoplankton and during the breakdown of marine algal cells are possible sources (Heikes et al., 2002; Nightingale, 1991; Sieburth and Keller, 1989). Recent laboratory culture experiments suggest that methanol is produced by a wide variety of phytoplankton including cyanobacteria (*Prochlorococcus marinus*, *Synechococcus* sp. and *Trichodesmium erythraeum*) and Eukarya (*Emiliania huxleyi*, *Phaeodactylum tricornutum* and *Nannochloropsis oculata, Dunaliella tertiolecta*) (Mincer and Aicher, 2016, Halsey et al., 2017). The mechanisms of *in situ* methanol production and their regulation remains largely unknown, although Halsey et al. (2017) reported light-dependent rates of methanol production in cultures of the marine green flagellate *Dunaliella tertiolecta* (cell size of 10−12 μm).

Methylotrophic bacteria are capable of utilising one-carbon compounds including methanol as their sole source of energy (methanol dissimilation) and carbon (methanol assimilation). Methylotrophs are widespread in terrestrial and aquatic systems (Kolb, 2009), but research into these bacteria in marine environments is still at an early stage. Traditionally, methylotrophs were thought to utilise methanol dehydrogenase (MDH encoded by *mxaF*, McDonald and Murrell, 1997) to metabolise methanol to formaldehyde, with further oxidation to $CO_2$ or incorporation of carbon into biomass (Chistoserdova, 2011; Chistoserdova et al., 2009). However, recent progress in this field has resulted in the discovery of the *xoxF* gene, encoding an alternative MDH (Wilson et al., 2008) and seemingly present in all known gram-negative methylotrophs to date (Chistoserdova, 2011;

Chistoserdova et al., 2009). The presence of methylotrophs in seawater has been confirmed using a range of molecular approaches including functional gene primers, stable isotope probing and metaproteomics (Dixon et al., 2013; Grob et al., 2015; Neufeld et al., 2008; Neufeld et al., 2007; Taubert et al., 2015). There are also bacterial cells that utilise methanol and other $C_1$ compounds for the production of energy but not biomass e.g. SAR11 for which Sun et al. (2011) proposed the new term 'methylovores', distinct from true methylotrophs which use $C_1$ compounds as sources of carbon and energy.

Limited studies of microbial methanol assimilation in the Atlantic Ocean have previously shown rates up to 0.42 nmol $l^{-1}$ $h^{-1}$ in recently upwelled coastal waters of the Mauritanian Upwelling (Dixon et al., 2013). However, open ocean waters of the Atlantic were substantially lower ranging between 0.002–0.028 nmol $l^{-1}$ $h^{-1}$ (Dixon et al., 2013). Microbial methanol dissimilation rates are generally up to 1000-fold higher than rates of assimilation; ranging between 0.70-11.2 and <0.001-0.026 nmol $l^{-1}$ $h^{-1}$ respectively for coastal waters (Sargeant et al., 2016; Dixon et al., 2011b). Methanol dissimilation rates ranging between 0.08–6.1 nmol $l^{-1}$ $h^{-1}$ have also been found in open ocean Atlantic waters (Dixon et al., 2011a). However, despite the ubiquity of methanol in seawater, the spatial extent or quantification of microbial methanol utilisation for energy production on a basin scale has not been previously investigated. Therefore, the objective of this research was to simultaneously characterise the spatial variability in microbial methanol dissimilation rates (at depths to 200 m) and in microbial community groups throughout contrasting biogeochemical regions of the Atlantic Ocean. This study represents the first basin-wide approach to investigating methanol as a source of reducing power and energy for microbes.

**2. Materials and Methods**

*2.1. Sampling strategy*

Sampling was carried out during an Atlantic Meridional Transect (AMT) (http://www.amt-uk.org). The research cruise (JC039, RRS James Cook, 13/10/09–01/12/09) departed from Falmouth, UK (50.15° N, 05.07° W) and arrived in Punta Arenas, Chile (53.14° S, 70.92° W). Water samples were collected daily from pre-dawn (97, 33, 14 and 1 % photosynthetically active radiation (PAR) equivalent depths and 200 m) and solar noon (97 %) conductivity-temperature-depth (CTD) casts. The PAR equivalent depths were 5 m, 10-31

m, 15-54 m and 38-127 m for the 97, 33, 14, 1 % light levels respectively and typically varied
with oceanic province. The pre-dawn and solar noon sampling periods were approximately
45-65 nautical miles apart (sampling locations are shown in Fig. 1). The Atlantic Ocean was
divided into five oceanic provinces, following the approach of Dixon et al. (2013), according
broadly to chlorophyll *a* concentrations (<0.15 mg m$^{-3}$ gyre regions, >0.15 mg m$^{-3}$ temperate
or upwelling regions, Fig. 1) with the northern gyre sub-divided into northern subtropical
gyre (NSG) and northern tropical gyre (NTG).   Measurements of the concentration of
methanol in seawater (Beale et al. 2013) and of methanol assimilation rates (Dixon et al.
2013) made during this transect have been reported previously.
*2.2. Microbial methanol uptake*
The oxidation of methanol to $CO_2$ (dissimilation) was determined using $^{14}$C-labelled
methanol (American Radiolabelled Chemicals Inc, Saint Louis, MO, USA) seawater
incubations as previously described in Dixon et al. (2011b). Seawater samples of 1 ml were
incubated with ~10 nM (final concentration) $^{14}$C-labelled methanol to measure rates of
microbial methanol dissimilation. Seawater methanol concentrations ranged between 48-361
nM (Beale et al., 2013) thus the radiotracer additions represent 3-21 % of *in situ*
concentrations in Atlantic waters. Incubations were conducted in triplicate, with 'killed'
controls (5 % trichloroacetic acid, TCA, final concentration), at *in situ* temperatures and in
the dark. Incubation temperatures were determined by the sea surface temperature recorded
by the corresponding CTD casts. Sample counts of $^{14}CO_2$, captured in the precipitate as
$Sr^{14}CO_3$ (nCi ml$^{-1}$ h$^{-1}$), were divided by the total $^{14}CH_3OH$ added to the sample (nCi ml$^{-1}$) to
calculate the apparent rate constants, $k$ (h$^{-1}$).
The incorporation of methanol carbon into microbial biomass (assimilation) was determined
using sample volumes of 320 ml to increase the total sample counts (Dixon et al., 2011b)
following procedures outlined in Dixon et al. (2011b, 2013). Filter sample counts were
divided by the total $^{14}CH_3OH$ added to the sample (nCi ml$^{-1}$) to calculate the apparent rate
constants, $k$ (h$^{-1}$). For both methanol assimilation and dissimilation, the specific activity of
$^{14}$C-labelled methanol (57.1 mCi mmol$^{-1}$) was multiplied by the apparent rate constants to
calculate rates of microbial methanol uptake (nmol l$^{-1}$ h$^{-1}$) following the approach of Dixon et
al. (2013). Evaluation of control samples suggests that ≤0.3 % of the added $^{14}CH_3OH$ is
recovered on the filters and ≤2 % in the resultant precipitate for methanol assimilation and
dissimilation respectively.
*2.3. Bacterial leucine incorporation*
Rates of bacterial leucine incorporation were measured using the incorporation of $^3$H-leucine
into bacterial protein in seawater samples using the method described by Smith and Azam
(1992). A final concentration of 25 nM (6.8 µl) of $^3$H-leucine (calculated using the specific
activity of 161 Ci mmol$^{-1}$, concentrations 1 mCi ml$^{-1}$, American Radiolabelled Chemicals Inc,
Saint Louis, MO, USA) was incubated with 1.7 ml seawater samples. Incubations were
conducted in triplicate with 'killed' controls (5 % TCA, final concentrations), at *in situ*
temperature and in the dark.
*2.4. Bacterial community composition*
Seawater samples of approximately twenty litres were collected for bacterial DNA analysis
from 97, 33, 1 and <1 % (200 m) PAR equivalent depths during pre-dawn CTD casts only.
Samples were filtered through 0.22 µm Sterivex polyethersulfone filters (Millipore, Watford,
UK) using a peristaltic pump. Filters were incubated with 1.6 ml of RNA Later (Life
Technologies, to preserve samples during shipment) overnight at 4° C, after which the RNA
Later was removed. Filters were stored immediately at -80° C before being shipped back to
the UK on dry ice and subsequently stored at -20 °C.
Bacterial DNA was extracted from filters using a modified phenol:chloroform:isoamyl
alcohol extraction method as previously described in Neufeld et al. (2007). Extracted DNA
was cleaned using Amicon ultra-0.5 centrifugal filter devices (Millipore) to remove any RNA
Later residue. The 16S rRNA gene primers 341F (Muyzer et al., 1993) and 907R (Muyzer et
al., 1998) were used for PCR amplification (32 cycles) with an annealing temperature of 55°
C. Purification of PCR products from agarose gels was conducted using the QIAquick gel
extraction kit (Qiagen, Crawley, UK) before being sent to Molecular Research LP (MR DNA,
http://www.mrdnalab.com) for 454 pyrosequencing using the GS-flx platform.
The 16S rRNA gene sequences were depleted of barcodes and primers, and then sequences
less than 200 bp, with ambiguous bases or with homopolymer runs exceeding 6 bp, were
removed. Sequences were de-noised and chimeras removed. After the removal of singleton
sequences, operational taxonomic units (OTUs) were defined at 97 % 16S rRNA gene
identity using Quantitative Insights Into Microbial Ecology (QIIME, http://qiime.org,
*Caporaso et al.* 2010). The OTUs were assigned taxonomically using BLASTn (Basic Local
Alignment Search Tool, NCBI) against the Silva database (http://www.arb-silva.de).
Sequences were randomly re-sampled to the lowest number of sequences per sample (386
sequences per DNA sample) to standardise the sequencing effort.
**3. Results**
*3.1. Microbial methanol dissimilation*
*3.1.1 Surface*
Pre-dawn surface rates of microbial methanol dissimilation ranged between 0.05–1.49 nmol l$^{-1}$
h$^{-1}$ throughout the transect of the Atlantic Ocean (Fig. 2a). Maximum variability in surface
rates of methanol dissimilation (average of 0.96 $\pm$ 0.45 nmol l$^{-1}$ h$^{-1}$, n=10) were observed
north of 25$^o$ N in NT and NSG regions. At the southern limit of the NSG, rates of methanol
dissimilation decreased sharply from 1.48 to 0.34 nmol l$^{-1}$ h$^{-1}$. Generally, surface rates
continued to decrease in a southward direction throughout the NTG and EQU regions,
reaching a minimum of 0.05 nmol l$^{-1}$ h$^{-1}$ in Equatorial upwelling waters. Interestingly, surface
rates started to gradually increase to 0.39 nmol l$^{-1}$ h$^{-1}$ in waters of the oligotrophic SG, before
declining to 0.18 nmol l$^{-1}$ h$^{-1}$ in the ST area. Methanol dissimilation rates determined at pre-
dawn (dark) generally exhibited a similar latitudinal pattern to those from solar noon (light).
Rates south of 25$^o$ N (NTG, EQU, SG, ST) showed a significant, almost 1:1 relationship,
between light (solar noon, y) and dark (pre-dawn, x) *in situ* sampling conditions (y=1.06x,
r=0.6240, n=13, P<0.05), with most variability between results from light versus dark
sampling occurring north of 25$^o$ N in NT and NSG provinces. This is most likely a reflection
of these waters exhibiting the greatest spatial variability, as the pre-dawn and midday stations
were typically 55 nautical miles apart.
*3.1.2 Depth distributions*
The average rates of methanol dissimilation with depth are shown in Fig. 3a for each oceanic
province. Rates varied between 0.05–1.68 nmol l$^{-1}$ h$^{-1}$, but showed no consistent statistically
significant trend with depth. However, clear differences were observed in microbial methanol
dissimilation in the top 200 m between contrasting provinces in the Atlantic Ocean; where
NSG≥NT>SG≈ST≥NTG>EQU. The highest rates of methanol dissimilation in the top 200 m
were observed in the most northern latitudes (0.22–1.50 and 0.15–1.68 nmol $l^{-1}$ $h^{-1}$ for NT
and NSG respectively), consistent with surface trends (Fig. 2a). A strong decrease was
observed between the NSG (0.99 ± 0.41 nmol $l^{-1}$ $h^{-1}$) and the NTG (0.18 ± 0.04 nmol $l^{-1}$ $h^{-1}$)
regions. However, rates of microbial methanol dissimilation determined in the oligotrophic
waters of the NTG (0.18 ± 0.04 nmol $l^{-1}$ $h^{-1}$) and SG (0.24 ± 0.12 nmol $l^{-1}$ $h^{-1}$) regions were
comparable with rates in the ST region (0.20 ± 0.05 nmol $l^{-1}$ $h^{-1}$), with the EQU exhibiting the
lowest average rates of 0.11 ± 0.03 nmol $l^{-1}$ $h^{-1}$.
Overall, latitudinal trends in depth profiles for methanol dissimilation rates mirrored those
found in surface waters. Surface microbial methanol dissimilation rates determined from pre-
dawn (x) water were compared to those from 200 m (y), which are in permanent darkness
(the deepest 1 % PAR equivalent depth of 175 m was found was in the SG at ~19.50ºS) and
also showed a ~1:1 relationship (y=0.967x, r=0.9237, n=19, P<0.001).
*3.2. Bacterial leucine incorporation rates*
*3.2.1 Surface*
Rates of bacterial leucine incorporation (BLI) varied between 2.9–25.2 pmol $l^{-1}$ $h^{-1}$ in the pre-
dawn surface waters of the Atlantic transect (Fig. 2b). On average, surface rates of BLI were
highest in the relatively more productive EQU upwelling region (18.3 ± 4.8 pmol $l^{-1}$ $h^{-1}$), and
lowest in the northern sub-tropical gyre (NSG, 5.2 ± 2.3 pmol $l^{-1}$ $h^{-1}$). Surface rates of BLI
averaged 7.8 ± 2.3 pmol $l^{-1}$ $h^{-1}$ and 7.7 ± 2.4 pmol $l^{-1}$ $h^{-1}$ in the NTG and SG regions
respectively. The one measurement of BLI in the ST suggested much higher rates (25.2 pmol
$l^{-1}$ $h^{-1}$) than previously determined during the transect, even when compared to the NT region
(9.9 ± 3.9 pmol $l^{-1}$ $h^{-1}$). Pre-dawn (dark) rates of BLI generally exhibited a similar latitudinal
pattern to those from solar noon (light), with more variability between light and dark
sampling observed in the waters of the productive EQU region. Bacterial rates of leucine
incorporation determined from samples collected at solar noon (y) were approximately 20%
less than those determined at pre-dawn (y=0.7815x, r=0.7288, n=22, P<0.001), perhaps
reflecting a degree of light inhibition of heterotrophic bacterial production.

## 3.2.2 Depth profiles

Rates of bacterial leucine incorporation varied between 0.5–60.2 pmol $l^{-1}$ $h^{-1}$ throughout the top 200 m of the water column. In the sunlit depths (97-1 % PAR) generally BLI rates followed the pattern EQY>NTG≈SG>NT>NSG (excluding the outliers of 60.2 and 31.3 pmol $l^{-1}$ $h^{-1}$ observed for the NSG at 14 % PAR from two depth profiles in this province). This trend differs slightly from that observed for surface only data due to sub-surface (1-14 % PAR) maxima observed in both the north and south oligotrophic gyres (NSG, NTG, SG). In the NT, NTG and EQU provinces, BLI rates were generally higher in sunlit depths compared to the dark at 200 m (Fig. 3b). However, there were no statistical differences between the provinces for rates of BLI determined at 200 m.

## 3.3. Bacterial community composition

### 3.3.1 Surface

The total number of operational taxonomic units (OTUs) sequenced throughout the Atlantic Ocean varied between 91–207. Overall, the largest contributors to surface bacterial communities were *Prochlorococcus* and SAR11 16S rRNA gene sequences (Fig. 5a); which together accounted for between 21-60 % of all OTUs (21% in the SG and 60% in the NSG). These bacteria typically numerically dominate surface waters of nutrient depleted oceanic regions e.g. Gomez-Pereira et al. (2013). The numbers of *Prochlorococcus*, determined via flow cytometry, for the same surface samples from which 16S rRNA genes were amplified range between 0.81 x $10^5$ for the NTG region and 3.10 x $10^5$ cells $ml^{-1}$ for the EQU region (see Table 2 for summary). *Prochlorococcus* 16S rRNA gene sequences contributed an average of 28 ± 12 % of the community composition of surface samples throughout the surface Atlantic Ocean. Numbers of SAR11 16S rRNA gene sequences contributed a maximum of 24 % to the total 16S rRNA gene sequences for the NSG region, and overall contributed an average of 11 ± 3 % to the bacterial community in surface waters of the Atlantic Ocean. There was a clear shift between surface bacterial communities in the two northern gyre provinces with *Prochlorococcus* and SAR11 16S rRNA gene sequences decreasing from the NSG to the NTG region (59 and 33 % of total 16S rRNA gene sequences respectively). *Oceanspirillales* and *Flavobacterales* 16S rRNA gene sequences contributed

approximately double the amount (compared to the total 16S rRNA sequences) in the NTG
compared to the NSG region (25 and 12 % respectively).
Microbial communities of the surface waters of the NT, NSG and EQU provinces were
dominated by *Prochlorococcus*, *Alteromonadales* and SAR11, together representing between
64–72 % of 16S rRNA gene sequences.  These orders were less dominant in the more
oligotrophic waters of the NTG and SG, accounting for 43 % and 34 % of 16S rRNA gene
sequences respectively. In these oligotrophic regions (NTG and SG) microbial communities
appear less dominated by a few orders, with a more even spread of bacterial orders
contributing to the community composition (Fig. 5a).
*3.3.2 Depth profiles*
The largest contributors to bacterial communities at the 33 % PAR depths were, like surface
communities, *Prochlorococcus* and SAR11 16S rRNA gene sequences (Fig. 5b). Together
they accounted for between 47-70 % of all OTUs, with the minimum and maximum
contributions in the SG and EQU provinces respectively. If the proportion of sequences
contributing individually <5% were included then collectively they accounted for between
69-91 % of all 16S rRNA gene sequences. The main differences between the surface and
33% PAR equivalent depth (14-31 m) are the increasing dominance of the cyanobacteria
*Prochlorococcus,* and the decrease in relative contribution of *Alteromonadales* at 33% PAR
depths, particularly in the NT region.
In the darker 1 % PAR depths (59-127 m) *Prochlorococcus* and SAR11 16S rRNA gene
sequences (Fig. 5c) still accounted for between 32-65 % of all OTUs, with the minimum and
maximum contributions in the SG and EQU respectively. With the addition of sequences for
each Order contributing <5 % to the total 16SrRNA gene sequences, these three categories
accounted for 60-81% of all 16S rRNA gene sequences retrieved throughout each of the
regions sampled. Two notable differences at this light level in the SG region compared to the
other provinces are the 12 % contribution made by the Order III *Incertae Sedis* which belongs
to the *Bacteroidetes* class, and the relative reduction in contribution made by
*Prochlorococcus* (11 % compared to an Atlantic average of 27±15 % at 1 % PAR). However,
the latter trend is not confirmed in the cell numbers of *Prochlorococcus* determined via flow
cytometry (Table 2).
In the permanent dark of 200 m, SAR11 bacteria contributed between 14-29 % in northern
regions, which contrasted to only 4-5 % in the EQU and SG provinces. The SAR324 clade
contributed 8-11 % in the northern gyre. Both uncultivated bacteria and those that
individually comprised <5 % contributed relatively highly to the OTUs (10-36 % and 21-33
% respectively). These two groupings together with the SAR11 and SAR324 make up 83-89
% in northern regions and between 37-56 % in the SG and EQU provinces respectively. For
the EQU region the *Alteromonadales* order is also significant at 25 % (which collectively
comprise 81 % of all OTUs for EQU), whilst for the SG the cyanobacteria *Prochlorococcus*
and *Synechococcus* comprise 52 % (which collectively comprise 89 % of all OTUs for SG).
**4. Discussion**
*4.1. Basin scale variability in biological methanol uptake*
Maximum rates of methanol dissimilation in the Atlantic Ocean were recorded in the NSG
province at 33 % PAR light depth (25 m, 1.68 nmol $l^{-1}$ $h^{-1}$, Fig. 2 and Fig. 4a). An overview
of the variation in rates of methanol dissimilation to $CO_2$ throughout the top 200 m of the
water column in the Atlantic Ocean is shown in Fig. 4a, which illustrates sub-surface maxima
in northerly latitudes. However, no statistically significant differences were ~~calculated~~ found
between rates of methanol dissimilation in the euphotic zone (97–1 % PAR) compared to the
aphotic zone (samples from 200 m) in the NSG ($t_{NSG}$=2.63, $t_{20}$=2.85 for P<0.01), NTG ($t_{NTG}$=
0.02, $t_{12}$=3.05 for P<0.01), EQU ($t_{EQU}$=1.01, $t_{18}$=2.88 for P<0.01) and SG regions ($t_{SG}$=0.88,
$t_{19}$=2.88 for P<0.01). This is consistent with a previous study in the north east Atlantic Ocean,
which similarly reported no significant variability in methanol dissimilation rates with depth
(Dixon and Nightingale, 2012). Nevertheless, greater variability with depth was observed for
methanol dissimilation rates from the northern gyre ($F_{NSG}$=3.22 where $F_{3,17}$=3.20, P<=0.05
and $F_{NTG}$=5.14 where $F_{2,10}$=4.10, P<0.05). Variability in rates from the euphotic zone were
found to be significantly higher than those from 200 m in northern ($t_{NT}$=3.17, $t_{20}$=2.85 for
P<0.01) and southern temperate regions ($t_{ST}$=5.03, $t_{10}$=3.17 for P<0.01).
Although the highest rates of methanol dissimilation were determined in the NSG, these
values were approximately seven times lower than the maxima determined during a seasonal
study of the temperate western English Channel (0.5-11.2 nmol $l^{-1}$ $h^{-1}$, Sargeant et al., 2016).
Rates determined in the temperate waters of the south Atlantic (0.11–0.45 nmol $l^{-1}$ $h^{-1}$) are
most comparable to the lowest rates determined during late spring and early summer of ~0.50
nmol l$^{-1}$ h$^{-1}$ in temperate northern coastal waters (Sargeant et al., 2016). The seasonal study
in the western English Channel showed maximum rates of up to 11.2 nmol l$^{-1}$ h$^{-1}$ during
autumn and winter months (Sargeant et al., 2016). The differences in methanol dissimilation
rates between the temperate waters of the North (0.83±0.42 nmol l$^{-1}$ h$^{-1}$) and South
(0.27±0.13 nmol l$^{-1}$ h$^{-1}$) Atlantic may therefore reflect seasonal differences between
hemispheres i.e. sampling in the NT region occurred during late autumn compared to late
spring in the ST region.
Methanol assimilation rates were generally two orders of magnitude lower than dissimilation
rates, reaching a maximum of 0.028 nmol l$^{-1}$ h$^{-1}$ in the top 200m throughout the Atlantic
Ocean (Fig. 4b). Rates of methanol assimilation exhibited sub-surface maxima (at 33% PAR
equivalent depth) which were particularly evident just north of the Equator (EQU) and in the
northern gyre (NSG) of 0.015±0.004 nmol l$^{-1}$ h$^{-1}$. These subsurface rates were on average
higher than surface values (0.004±0.004 nmol l$^{-1}$ h$^{-1}$). Results are similar to findings by
Dixon and Nightingale (2012) who also demonstrated sub-surface maxima between 20–30 m
in the north east Atlantic. The methanol assimilation rates are shown for direct comparison to
dissimilation, but have been previously discussed in more detail in Dixon et al. (2012).
*4.2. Bacterial community and productivity*
In contrast to microbial methanol dissimilation, rates of bacterial leucine incorporation were
lowest in the northern oligotrophic gyre (NSG 5.2 ± 2.3 pmol l$^{-1}$ h$^{-1}$, NTG 7.8 ± 2.3 pmol l$^{-1}$
h$^{-1}$) reflecting lower microbial activity in these regions of the Atlantic. Surface microbial
methanol dissimilation rates exhibited a statistically significant inverse correlation with
bacterial leucine incorporation, ($r$ = -0.351, $n$ = 36, $P$ ≤ 0.05). This is consistent with
findings from a seasonal study in the western English Channel, where surface rates of
methanol dissimilation were also inversely correlated to bacterial production (Sargeant et al.,
2016). For all the depth data a negative correlation was also found in the NTG, EQU and SG
regions ($r$ = -0.372, $n$ = 52, $P$ ≤ 0.01), but NT and NSG areas showed methanol dissimilation
rates independent of BLI. The productivity of heterotrophic bacteria is generally associated
with the concentrations of phytoplankton-derived dissolved organic matter (DOM) e.g.
proteins, lipids and carbohydrates which are utilised as sources of energy and carbon (Benner
and Herndl, 2011; Nagata, 2008; Ogawa and Tanoue, 2003). Results from this present study
indicate that in regions of low heterotrophic bacterial production i.e. in the northern Atlantic
Gyre (minimum rate of bacterial leucine incorporation of 3 pmol $l^{-1}$ $h^{-1}$) rates of methanol
dissimilation were relatively higher. In oligotrophic regions, phytoplankton-derived DOM is
scarce, suggesting that those bacteria able to metabolise methanol are using the carbon from
methanol as an alternative source of energy (and to a lesser extent carbon).
Although the bacterial community 16S rRNA gene sequence data did not display any clear
patterns with changing biogeochemical province (in contrast to microbial methanol
dissimilation rates), the bacterial community was shown to be depth-stratified throughout the
Atlantic Ocean (Fig. 6a). A non-metric multi-dimensional scale (MDS) plot of a Bray-Curtis
similarity matrix of 16S rRNA gene sequences (Fig. 6a) found bacterial community samples
to cluster into three distinct groupings possibly reflecting light levels: sunlit (97 and 33 %
PAR), minimal light (1 % PAR) and dark (200 m). Bacterial community samples from the
same PAR equivalent depths were found to group together regardless of biogeochemical
province. A larger cluster formed of samples from 97 and 33% PAR is likely to be formed of
bacterial communities originating from the well-mixed surface layer of the water column,
accounting for their similarity in composition. When all environmental parameters were
considered together (including bacterial numbers and BLI) a Euclidean distance matrix non-
metric MDS also demonstrated photic waters (97-1 % PAR) clustered together, and were
significantly different to dark waters from 200m (Fig. 6b). However, no significant
differences were observed between rates of methanol dissimilation determined from the
euphotic zone (samples from 97-1 % PAR equivalent depths) compared to the aphotic zone
(samples from 200 m depth) for gyre and equatorial regions (NSG $t_{NSG}$=2.63 ($t_{20}$=2.85 for
P<0.01), NTG $t_{NTG}$=0.02 ($t_{12}$=3.05 for P<0.01), EQU $t_{EQU}$=1.01 ($t_{18}$=2.88 for P<0.01) and SG
$t_{SG}$=0.88 ($t_{19}$=2.88 for P<0.01)) although, clear differences between provinces were evident
(Fig. 6c). This is consistent with results from Dixon and Nightingale (2012) who also found
no significant variation of methanol dissimilation with depth in the north east Atlantic Ocean.
These data suggest that light levels do not have a strong role to play in microbial methanol
dissimilation in waters of the Atlantic, despite the overall bacterial community showing
strong variability with depth (or incident light). Depth-stratification of microbial communities
has been observed previously by Carlson et al. [2004], DeLong et al. [2006] and between
euphotic and aphotic zones in the north western Sargasso Sea (Carlson et al., 2004).
Heywood et al. (2006) suggested that the physical separation of low nutrient surface waters in
gyre regions from mixing with more nutrient rich waters below a defined pycnocline, in
combination with differing levels of light availability, could partially explain changes in
bacterial community composition throughout the water column. Therefore, these results could
indicate that methanol dissimilation is limited to specific microbial groups that are present
relatively uniformly between the surface and 200m, although more depth variability is shown
north of 25°N where rates of methanol dissimilation are the highest and most variable.
*4.3. Methanol dissimilation and SAR11*
SAR11 cells have been shown to utilise methanol, but only as a source of energy (Sun et al.,
2011). The numbers of SAR11 16S rRNA gene sequences exhibited a statistically significant
correlation with rates of microbial methanol dissimilation throughout the Atlantic basin ($r =$
0.477, $n = 20$, $P < 0.05$), where the number of SAR11 16S rRNA gene sequences explained
approximately half of the spatial variability in rates of methanol dissimilation. It should be
noted that this correlation has been made with amplicon ratios, relating to the relative success
of SAR11 in the community, rather than with SAR11 cell numbers specifically. In culture,
SAR11 cells (strain HTCC1062) have previously been shown to utilise methanol as a source
of energy at a rate of ~5 x $10^{-20}$ moles cell$^{-1}$ h$^{-1}$ (Sun et al., 2011), which equates to 2 nmol l$^{-1}$
h$^{-1}$ (using a culture cell abundance of 4 x $10^7$ cells mL$^{-1}$, Sun et al., 2011). SAR11 cells
dominate (59 ± 4%) the low nucleic acid (LNA) fraction of bacterioplankton consistently
across the Atlantic Ocean, where typically numbers of LNA range between 0.2-1.0 x $10^9$ cells
l$^{-1}$ (Mary et al., 2006a). Thus estimates of *in situ* SAR11 numbers range between 0.12-0.59 x
$10^9$ cells l$^{-1}$. This is consistent with estimates from the Sargasso Sea of ~0.1 x $10^9$ cells l$^{-1}$
(where they are reported to contribute ~25% of total prokaryotic abundance of 0.4 x $10^6$ cells
mLl$^{-1}$, Malmstrom et al., 2004). Thus, we estimate that SAR11 cells of the Atlantic Ocean
could be oxidising methanol at rates between 5-29.5 pmol l$^{-1}$ h$^{-1}$, which could account for
between 0.3-59 % of the rates of methanol dissimilation in surface Atlantic waters.
A seasonal investigation in the western English Channel reported bacterial numbers ranging
between 2.0-15.8 x$10^5$ cells ml$^{-1}$ (Sargeant et al., 2016) which agrees well with data from
Mary et al. (2006b, 2.0-16.0 x$10^5$ cells ml$^{-1}$). Assuming that SAR11 contribute between 9-
20% of total bacterioplankton (Mary et al., 2006b) suggests SAR11 numbers range between
0.18-3.16 x$10^5$ cells ml$^{-1}$ at this coastal site. Using the above estimate of ~5 x $10^{-20}$ moles cell$^{-}$
$^1$ h$^{-1}$ for rates of methanol dissimilation in cultured SAR11 cells suggests that SAR11 could
oxidise methanol at rates ranging between 0.9-15.8 pmol l$^{-1}$ h$^{-1}$ in temperate coastal regions.
This equates to <0.01-2.3% of microbial community methanol dissimilation rates (0.7-11.2
nmol l$^{-1}$ h$^{-1}$, Sargeant et al., 2016). Therefore, we suggest that cells of the SAR11 clade are
more likely to make a larger contribution to marine microbial methanol dissimilation in open
ocean environments, where alternative sources of carbon are more limited relative to
temperate coastal waters.  More work is required to add clarity and understanding to the role
that SAR11 cells play in marine community methanol dissimilation.
Previously methylotrophic bacteria such as *Methylophaga* sp., *Methylococcaceae* sp. and
*Hyphomicrobium* sp. have been identified, using *mxaF* functional gene primers (which
encode for the classical methanol dehydrogenase), from the same DNA samples analysed for
16S rRNA genes in this study, from the upper water column of Atlantic Ocean provinces
Methylotrophic bacteria such as *Methylophaga sp.*, *Methylococcaceae sp.* and
*Hyphomicrobium sp.* have been previously identified, using *mxaF* functional gene primers
(which encode for the classical methanol dehydrogenase), from the upper water column of
Atlantic Ocean provinces (Dixon et al., 2013). Although numerically very rare (1-11 16S
rRNA gene sequences per sample), 16S rRNA gene sequences identified as *Methylophaga*
spp., *Methylophaga* sp. DMS021 (EU001861) and uncultured *Methylophaga* sp. (EU031899),
were found in each of the Atlantic Ocean provinces in this study (at 97% PAR or 200m
depth), consistent with previous identification of *Methlophaga* spp. in these Atlantic
provinces using *mxaF* gene cloning in (Dixon et al., 2013).  More recently the *xXoxF* gene,
which encodes an alternative methanol dehydrogenase, has also been found to be widespread
in coastal marine environments (Taubert et al., 2015). SAR11 bacteria are thought to contain
an Fe alcohol dehydrogenase, which although not specific for methanol, can oxidise methanol
(and other short chain alcohols) to formaldehyde which is then thought to be converted to
CO$_2$ by a methyl-THF linked oxidation pathway tetrahydrofolate-linked C1 transfer pathway
to produce energy (Sun et al., 2012). Thus it seems likely that both methylotrophic bacteria
possessing *mxaF* and/or *xoxF,* together with microbes such as SAR11 (Sun et al., 2011), are
largely responsible for the turnover of methanol in seawater.
Members of *Betaproteobacteria*, OM43, have been shown to be potentially important
obligate methylotrophs, with cultivated cells of strain HTCC2181 dissimilating 3.5 times
more methanol than was assimilated (Halsey et al., 2012).  OM43 were not successfully
identified in the 16S rRNA sequences in this study, which could be an artefact of the
relatively low sequence coverage (386 sequences per sample) leading to this taxon not being

detectable. During a previous coastal study, also analysing 16S rRNA pyrosequence data, in the western English Channel (Sargeant et al., 2016) only a single sequence of the OM43 clade, HTCC2181, was identified. This is a limitation of this type of environmental sequencing effort and should be a consideration in planning any future projects aiming to understand microbial function through process measurements alongside the generation of metagenomic datasets."

*4.4. Marine methanol cycling*

Data from this study substantially add to the measurements of microbial methanol dissimilation rates in seawater. This extended spatial coverage clearly demonstrates that methanol dissimilation is a widespread microbial process taking place in light and dark environments throughout the Atlantic Ocean. Dissimilation rates are typically two orders of magnitude greater than assimilation rates across most of the Atlantic Basin. These data suggest that methanol is an important source of energy for microbes. This is particularly true in the northern oligotrophic waters of the Atlantic Ocean, where corresponding *in situ* methanol concentrations range between 148-281 nM (Table 1). What is not clear is the source of methanol in open ocean waters, which is suspected to be biological in nature (Dixon et al., 2011a). Although direct flux estimates suggest that the atmosphere could also act as a source to the ocean (Yang et al, 2013), the magnitude of this flux is insufficient to support the observed rates of microbial methanol consumed by bacteria, and hence is suspected to be a minor contribution (Dixon et al., 2011a). Recent culture studies indicate that *Prochlorococcus sp.*, *Synechococcus* sp. and *Trichodesmium sp.* could produce methanol (Mincer and Aicher, 2016, Halsey et al., 2017), but *in situ* production mechanisms are unknown. Further work is needed to fully elucidate and quantify the sources of methanol in marine waters.

**5. Conclusions**

This study reports the first basin-wide understanding of microbial methanol dissimilation rates in seawater. Radiochemical assays have demonstrated active metabolism throughout the top 200 m of the water column, with rates being substantially higher in the northern subtropical Atlantic gyre. Microbial methanol dissimilation rates showed a positive

correlation with the numbers of SAR11 16S rRNA gene sequences, and an inverse relationship with bacterial leucine incorporation. Future work should determine marine methanol sources and understand the relative contribution of various microbial orders to methanol loss processes.

**Acknowledgements**

We thank all scientists, officers and crew of the RRS James Cook during JC029. We also thank Rachael Beale for determining methanol concentrations, Glen Tarran for flow cytometry data, Karen Tait, Yin Chen and Michael Cunliffe for advice with molecular biology work. Satellite data were processed by the NERC Earth Observation Data Acquisition and Analysis Service (NEODAAS) at Plymouth Marine Laboratory (http://www.neodaas.ac.uk). This work was funded by the UK Natural Environmental Research Council (NERC) and the Earth and Life Systems Alliance, Norwich Research Park. This study is also a contribution to the international IMBeR project and was also supported by the UK Natural Environment Research Council National Capability funding to Plymouth Marine Laboratory and the National Oceanography Centre, Southampton. This is contribution number 323 of the AMT programme.

**Conflict of Interest Statement**

The Authors declare no conflict of interest with this manuscript.

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

**Figure and Table legends.**
**Figure 1.** Remotely sensed MODIS-Aqua chlorophyll *a* composite image of the Atlantic
Ocean from November 2009 (image courtesy of NEODAAS). White squares represent
sampling points and ~~circles~~ ovals indicate samples within different oceanic provinces,
labelled with province names NT (northern temperate), NSG (northern subtropical gyre),
NTG (northern tropical gyre), EQU (equatorial upwelling), SG (southern gyre), ST (southern
temperate).
**Figure 2**. Variability in a) microbial methanol dissimilation rates (using the specific activity
of $^{14}CH_3OH$) and b) bacterial leucine incorporation (BLI), in surface waters of the Atlantic
Ocean. Rates were determined from pre-dawn (♦ solid line) and solar noon (◊ dashed line)
CTD casts. Error bars represent ±1 s.d. of triplicate samples, dashed vertical lines indicate
Atlantic province boundaries.
**Figure 3.** Average depth profiles in Atlantic provinces for a) microbial methanol
dissimilation (using the specific activity of $^{14}CH_3OH$) and b) bacterial leucine incorporation
(BLI) in pre-dawn waters. Error bars represent ±1 s.d. of variation within the province,
province averages derived from NT (n = 5), NSG (n = 5), NTG (n = 3), EQU (n = 4), SG (n =
5) and ST (n = 3), except for BLI where there is no data from the ST.
**Figure 4.** Microbial methanol (a) dissimilation and (b) assimilation rates (nmol l$^{-1}$ h$^{-1}$) in the
top 200 m of an Atlantic Meridional transect (contour plots).
**Figure 5.** Changes in bacterial community composition (Order, identified using 16S rRNA
gene sequencing) for a) 97 % PAR surface 5m, b) 33 % PAR 10-31m, c) 1 % PAR 15-54m
and d) 200 m for different provinces (NT, NSG, NTG, EQU and SG) of the Atlantic Ocean.
Analysis is based on a rarefied sample of 386 sequences per sample. Bacterial Orders
individually contributing to less than 5% of the total sample sequences were pooled together
into 'Others (<5%)' for clarity. Where ▬ *Prochlorococcus*, ▬ *Alteromonadales*, ▭
SAR11 clade, ▬ *Oceanospirillales*, ▬ *Rhodospirillales*, ▬ *Flavobacteriales*, ▬
*Rhodobacterales*, ▭ *Sphingomonadales*, ▨ *Synechococcus*, ▭ *Acidimicrobiales*, ▭
Order III *Incertae Sedis*, ▨ SAR324 clade (Marine group B), ▬ uncultivated bacterium, ▬
other bacteria individually comprising <5%.
**Figure 6.** Non-metric multi-dimensional scale (MDS) plots of (a) a Bray-Curtis similarity
matrix of the 16S rRNA gene sequences of the bacterial community, (b) a Euclidean distance
matrix of environmental parameters (salinity, temperature, chl. a, primary productivity,
inorganic nutrients, flow cytometry cell numbers, BLI) and (c) a Euclidean distance matrix of
rates of methanol dissimilation. Dashed lines highlight significant sample grouping. Plots
generated using PRIMER-E (www.primer-e.com). For (a) and (b) ■ represents samples from
200 m i.e. 0 % PAR.
**Table 1.** Summary of rates of methanol uptake (dissimilation and assimilation), methanol
concentrations, bacterial leucine incorporation (BLI) and production (BP), numbers of
heterotrophic bacteria (BN), *Prochlorococcus* (Pros) and *Synechococcus* (Syns). "Values
given are average ± standard deviation (range). NA denotes that data is not available."

**Figure 1.**

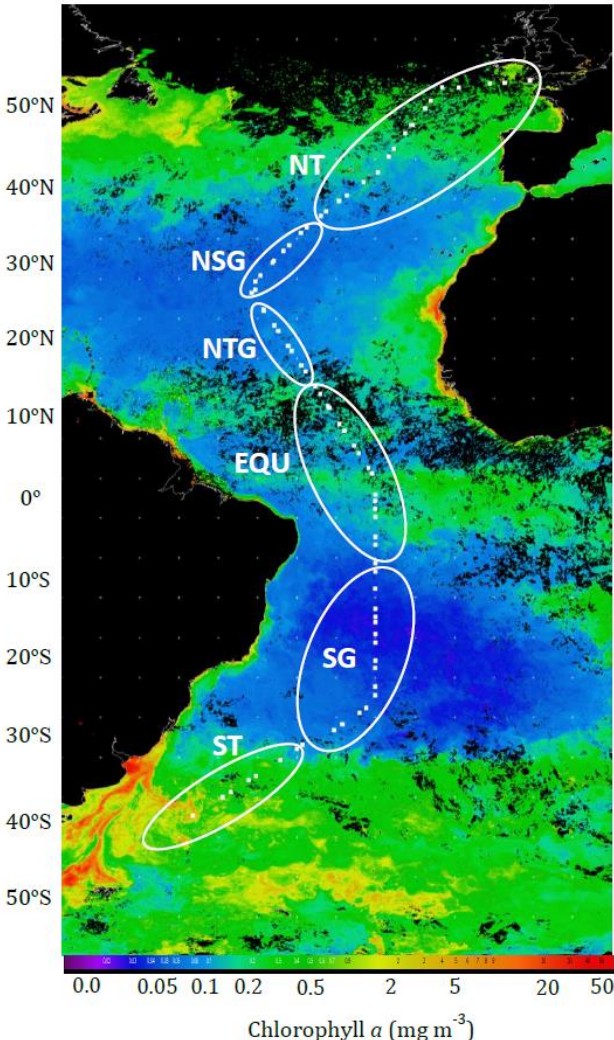

**Figure 2.**

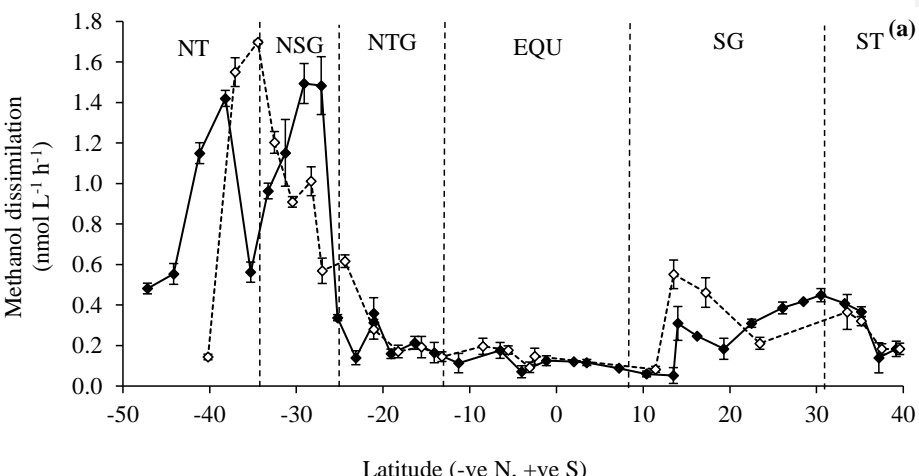

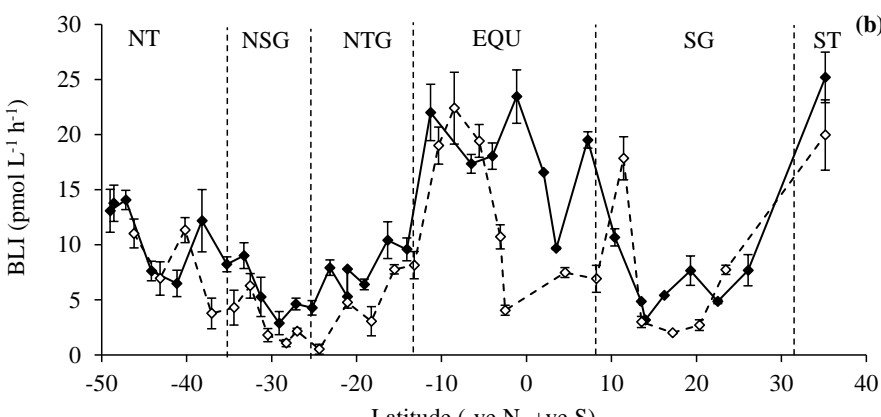

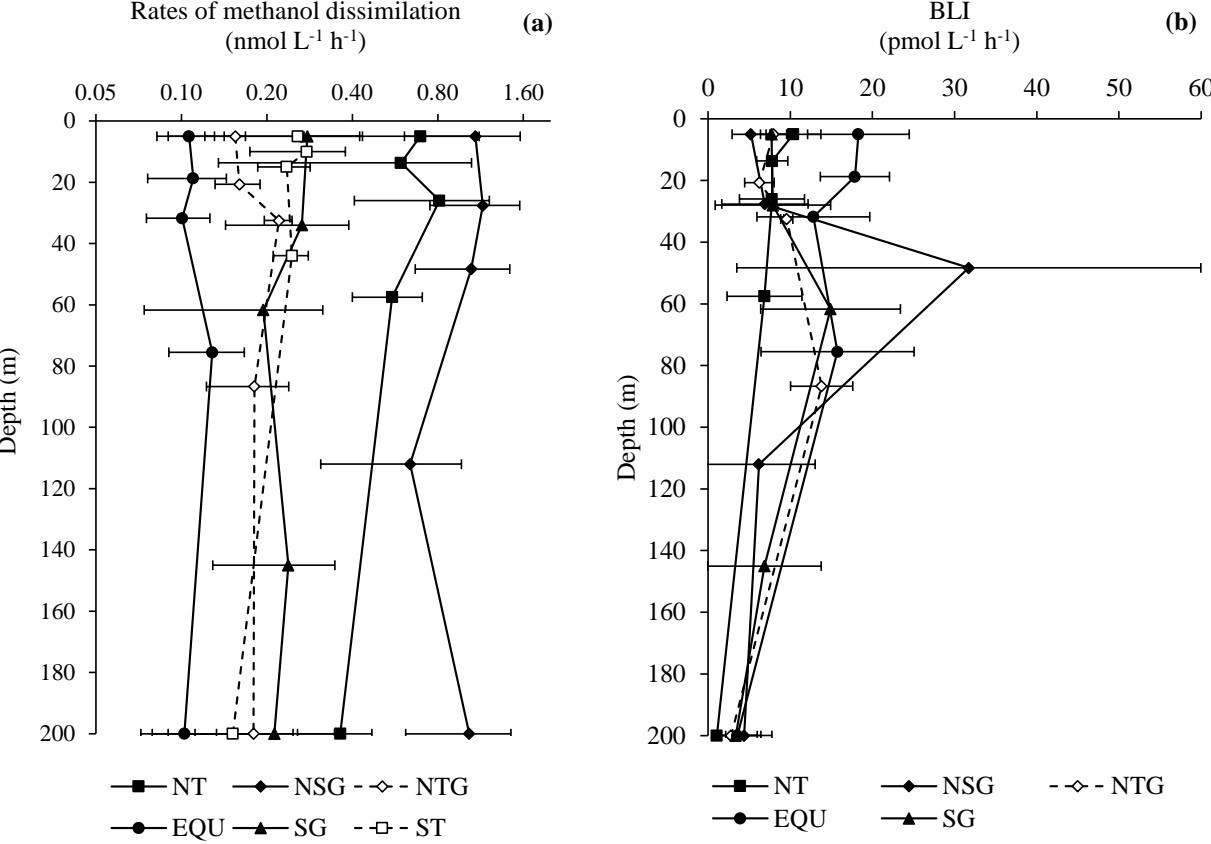

**Figure 3.**

**Figure 4.**

(a)

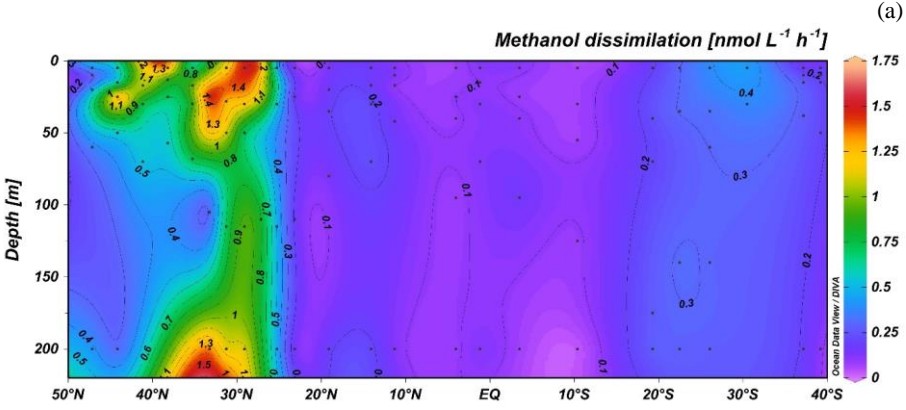

(b)

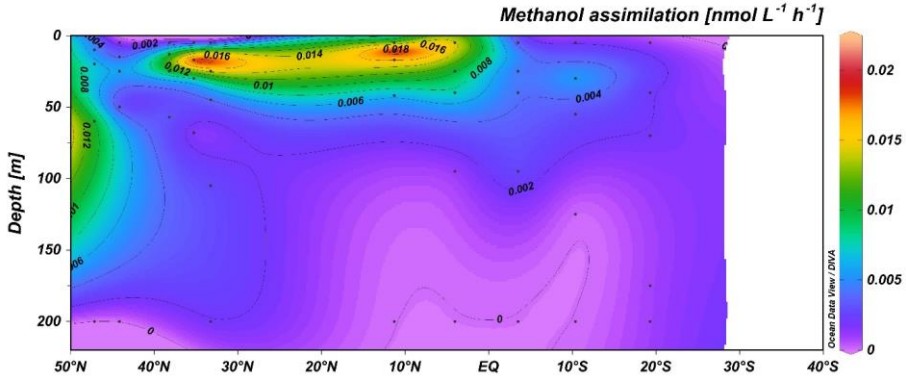

**Figure 5.**

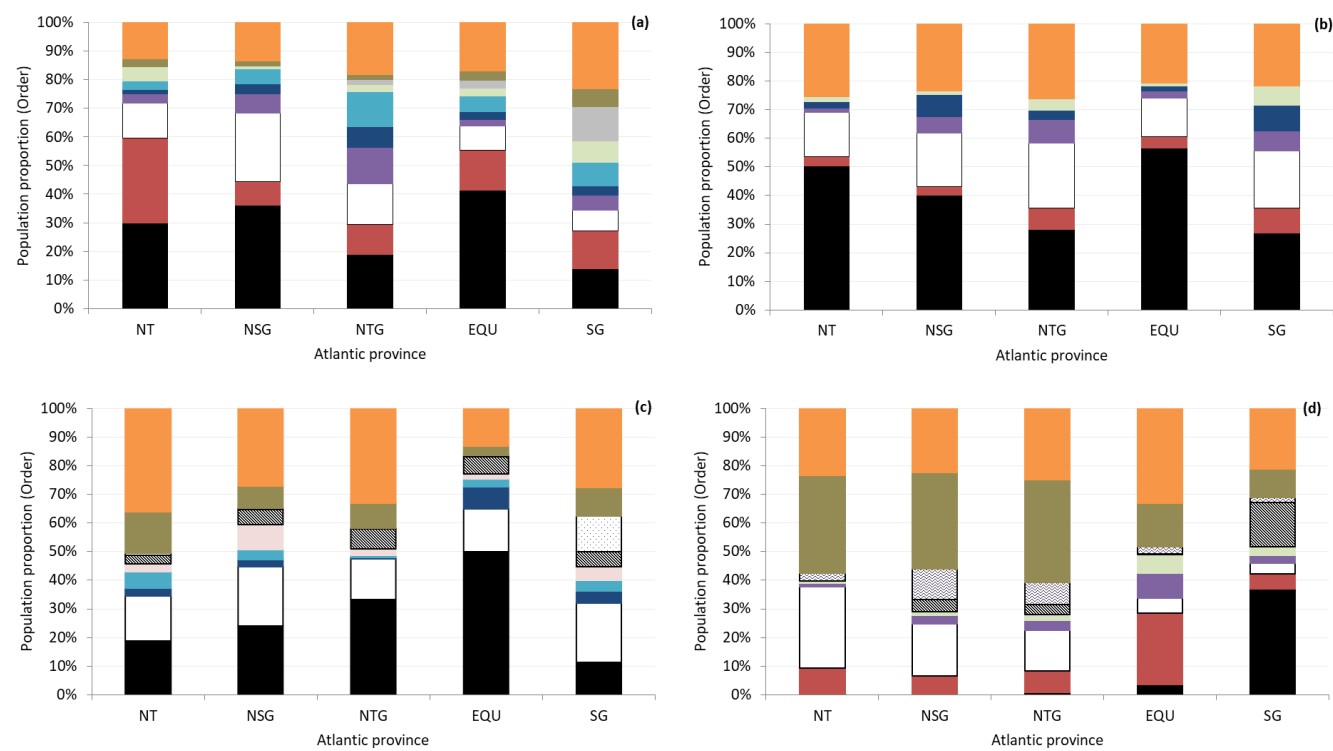

**Figure 6.**

(a)

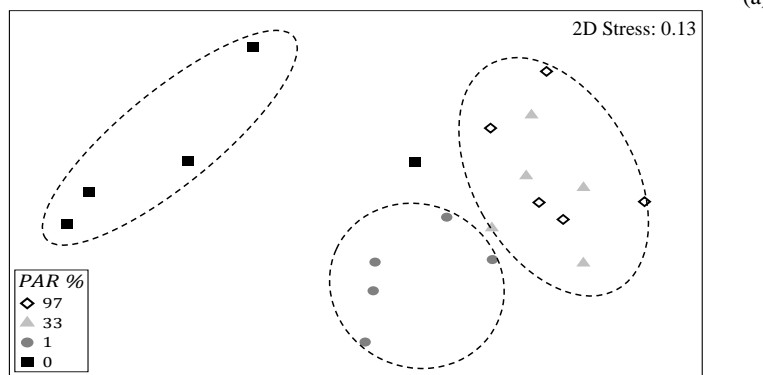

(b)

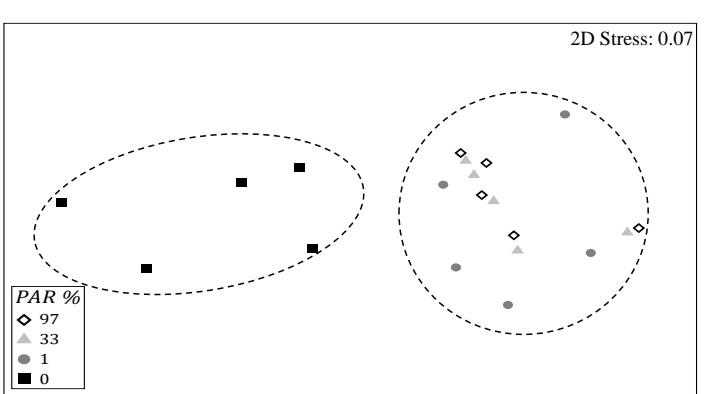

(c)

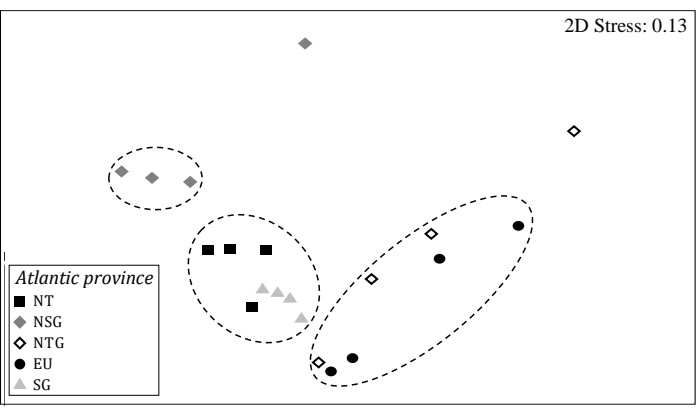

**Table 1.** Summary of rates of methanol uptake (dissimilation and assimilation), methanol concentrations, bacterial leucine incorporation (BLI) and production (BP), numbers of heterotrophic bacteria (BN), *Prochlorococcus* (Pros) and *Synechococcus* (Syns). Values given are average ± standard deviation (range). NA denotes that data is not available.

| | | Atlantic province | | | | | |
|---|---|---|---|---|---|---|---|
| | **Overall** | **NT** | **NSG** | **NTG** | **EQU** | **SG** | **ST** |
| Methanol dissimilation ($nmol\ L^{-1}\ h^{-1}$) | 0.45±0.42 (0.01–1.68) | 0.69±0.35 (0.22–1.50) | 0.99±0.41 (0.15–1.68) | 0.18±0.04 (0.10–0.25) | 0.11±0.03 (0.07–0.17) | 0.24±0.12 (0.01–0.45) | 0.20±0.05 (0.11–0.27) |
| Methanol assimilation (x $10^{-2}$) ($nmol\ L^{-1}\ h^{-1}$) | 0.51±0.54 (0.00–2.24) | 0.54±0.53 (0.00–2.23) | 0.53±0.56 (0.17–1.51) | NA | 0.67±0.66 (0.00–2.24) | 0.19±0.16 (0.00–0.57) | NA |
| BLI ($pmol\ L^{-1}\ h^{-1}$) | 9.4±8.9 (0.5–60.2) | 7.7±4.0 (0.9–14.2) | 9.7±14.2 (1.0–60.2) | 8.0±4.3 (2.0–17.0) | 13.7±7.9 (0.6–26.4) | 8.2±9.5 (0.5–41.5) | NA |
| [a]BP(TCF) ($ng\ C\ L^{-1}\ h^{-1}$) | 14.6±13.8 (0.8–96.1) | 11.9±6.1 (1.5–22.0) | 15.0±21.9 (1.5–96.1) | 12.4±6.6 (3.2–26.3) | 21.2±12.2 (1.0–41.0) | 12.7±14.8 (0.8–64.3) | NA |
| [b]BP (ECF) ($ng\ C\ L^{-1}\ h^{-1}$) | 4.8±4.6 (0.3–31.6) | 3.9±2.0 (0.5–7.2) | 4.9±7.2 (0.5–31.6) | 4.1±2.2 (1.0–8.7) | 7.0±4.0 (0.3–13.5) | 4.2±4.9 (0.3–21.1) | NA |
| Numbers of heterotrophic bacteria ($x10^5$) ($cells\ mL^{-1}$) | 6.5±6.3 (1.4–82.6) | NA | NA | 5.8±2.0 (1.6–9.8) | 8.8±10.3 (1.4–82.6) | 5.4±4.4 (1.5–35.8) | NA |
| Numbers of *Prochlorococcus* sp. ($x10^5\ cells\ mL^{-1}$) | 1.12±4.62 (0.0-4.62) | 0.91±0.07 (0.0-2.56) | 0.89±0.71 (0.0-4.21) | 1.52±1.23 (0.0-4.19) | 1.67±0.2 (0.0-4.62) | 1.20±0.01 (0.0-2.45) | 0.35±0.22 (0.0-2.33) |
| Numbers of *Synechococcus* sp. ($x10^4\ cells\ mL^{-1}$) | 1.64±31.4 (0.0-31.4) | 1.96±3.61 (0.0-12.7) | 0.18±0.21 (0.0-0.93) | 0.15±0.17 (0.0-0.73) | 1.34±2.69 (0.0-12.8) | 0.14±0.13 (0.0-0.79) | 8.30±10.3 (0.02-31.4) |
| [c]Methanol (nM) | 143±82 (38–420) | 110±126 (38–420) | 203±38 (154–281) | 193±46 (148–278) | 148±37 (117–241) | 110±33 (58–176) | 132 |

[a]Theoretical conversion factor (TCF) 1.55 kg C mol leu$^{-1}$, [b]empirical conversion factor (ECF) 0.51 kg C mol leu$^{-1}$, [c]From Beale et al., (2013)