# Peer review of "MS No.: bg-2018-30 Title: Basin-scale variability of microbial methanol uptake in the Atlantic Ocean"

_Biogeosciences, 2018_

## Referee Comment (RC1) · L. Chistoserdova (Referee) · 8 Mar 2018

I already posted my comments and have no further comments

---

## Short Comment (SC1) · 8 Mar 2018

This study provides an inventory of measurements relevant to methanol consumption by microbial communities across the Atlantic, a rare basin-wide evaluation. The description is somewhat monotonous, but it is what it is. A great variability is uncovered across provinces and across depths, but little correlation is found of methanol oxidation/assimilation with respect to where it happens. In general, dissimilation is somewhat correlated with the presence of SAR11, and, in general, assimilation is two orders of magnitude lower than dissimilation. Which means SAR11 uses some other carbon source(s) for building biomass, and these remain unknown. In general, I think, even if many questions remain unanswered, this is a useful benchmark study.

Improvements that I would like to suggest:

Page 14, line 15, please say tetrahydrofolate-linked C1 transfer pathway, there are various oxidation levels and none of them are methyl- level after methanol oxidation.

Same page, lines 15-18. You do not see any bona fide methylotrophs in your 16S libraries. How can you conclude that they are active, along with SAR11? Either elaborate or remove this statement. PCR amplification of specific genes does not compare with 16S analysis, and you do not do any in this study anyway.

Meantime, an interesting question: while true methylotrophs do inhabit marine waters, why are they so sparse and apparently uncompetitive compared to SAR11? Can you elaborate?

Fig. 5 would greatly benefit from introducing colors, would be so much easier to compare guild distribution. Also, please order the taxa in a uniform way, i.e. use the same taxon order in each panel.

Table 1. Specify that you show ranges below averages/means, specify which. Specify what NA means.

---

## Author Comment (AC1) · 8 Mar 2018

Thank your for your valued comments and thoughts. We will be sure to address them in a revised manucript.

---

## Referee Comment (RC2) · S. Giovannoni (Referee) · 11 Mar 2018

This is a valuable study that adds significantly to our understanding of methanol oxidation in the oceans. The authors report seawater methanol oxidation rates obtained with 14C tracer methods, and microbial diversity measurements, from a latitudinal transect between 40S and 50N. They find that methanol oxidation rates are correlated with SAR11 relative abudance. Overall, the reported rates of methanol oxidation are in good agreement with previous measurements, but this study is exceptional in geographical scope and exploration of variables such as community composition and depth. Interestingly, the manuscript reports an inverse correlation between bacterial production estimated by the 3H leucine method and methanol oxidation. Although there have been a number of reports previously on methanol cycling in the oceans, I see the sub-

ject coming of age with this report, which confirms what we knew and also shows us new trends that could only have been observed by making extensive measurements across a latitudinal transect.

A couple of comments follow about aspects of the paper that could be improved.

1. I recommend commenting on the abundance of methylophaga and OM43 in the 454 data, or indicate they were not detected if that is the case. It may be that the relatively low coverage obtained in this study (386 seqs/sample) led to these taxa being undetectable. If this is the case, that should be explained so that readers new to this topic understand the issues. OM43 is not mentioned at all, but perhaps it should be, since it has been shown to be an obligate methylotroph, is one of the dominant taxa in some coastal environments, and has been shown to be a source of abundant XoxF peptides in a coastal ocean metaproteome.

2. Amplicon ratios are not as powerful as cell numbers for identifying correlations between taxa and rates, although they are much easier to obtain. So, the correlations with SAR11 are not with SAR11 cells per unit volume, which would be best, but rather a correlation between the relative success of SAR11 in the community and rates of MeOH oxidation. I suggest the authors revisit the manuscript and choose wording that conveys these issues to oceanographer readers, who often misunderstand this aspect of relative abundance data.

---

## Author Comment (AC2) · 15 Mar 2018

Many thanks for your very positive encourgaing review and comments. We will of course incorporate all your recomendations into a revised manucript.
* * *

---

## Author Comment (AC3) · 9 Apr 2018

Excellent, all your comments will be incorporated into a finalised manuscript, many thanks, Jo Dixon

---

## Author Comment (AC4) · 24 May 2018

Each of L. Chistoserdova's comments has been addressed individually as follows.

Page 14, line 15, please say tetrahydrofolate-linked C1 transfer pathway, there are various oxidation levels and none of them are methyl- level after methanol oxidation. - "methyl-THF linked oxidation pathway" will be changed to "tetrahydrofolate-linked C1 transfer pathway" (Page 14, line 15).

Same page, lines 15-18. You do not see any bona fide methylotrophs in your 16S libraries. How can you conclude that they are active, along with SAR11? Either elaborate or remove this statement. PCR amplification of specific genes does not compare with 16S analysis, and you do not do any in this study anyway. –

Specific gene amplification, using mxaF functional gene primers, has been conducted previously on the same samples as the current study looking at 16S rRNA analysis. The mxaF functional gene analysis identified classic methylotrophic bacteria from these samples, these results are published in a previous manuscript Dixon et al. (2013). To clarify this in the text we propose the amendment of "Methylotrophic bacteria such as Methylophaga sp., Methylococcaceae sp. and Hyphomicrobium sp. have been previously identified, using mxaF functional gene primers (which encode for the classical methanol dehydrogenase), from the upper water column of Atlantic Ocean provinces (Dixon et al., 2013)" to "Previously methylotrophic bacteria such as Methylophaga sp., Methylococcaceae sp. and Hyphomicrobium sp. have been identified, using mxaF functional gene primers (which encode for the classical methanol dehydrogenase), from the same DNA samples analysed for 16S rRNA genes in this study, from the upper water column of Atlantic Ocean provinces (Dixon et al., 2013)" (Page 14, lines 7-10).

Meantime, an interesting question: while true methylotrophs do inhabit marine waters, why are they so sparse and apparently uncompetitive compared to SAR11? Can you elaborate? –

The authors agree this is an interesting question and more work is needed to unpick this. We don't have an answer for this based on published literature and knowledge currently available, however we can speculate that it may be down to the shear abundance and evolutionary strategy of SAR11 in comparison to true methylotrophs. SAR11 are the most abundant, free living, heterotrophic bacteria in open ocean systems and are often the most abundant organisms in oligotrophic waters. The competitiveness and high abundance of SAR11 cells in open ocean waters could be one part of a reason why true methylotrophs are relatively sparse in comparison. SAR11 have been shown to have one of the smallest genome sizes of any replicating cell and Giovannoni et al., (2005) suggest that the streamlining hypothesis may provide an explanation for this. The streamline hypothesis, assumption that selection reduces genome size due to the

metabolic burden of replicating DNA without adaptive value, could be the strategy responsible for the dominance and success of the SAR11 clade in oligotrophic waters.

Fig. 5 would greatly benefit from introducing colors, would be so much easier to compare guild distribution. Also, please order the taxa in a uniform way, i.e. use the same taxon order in each panel. –

We will change shading of Figure 5 to coloured taxa for clarification and amend the order of taxa presented within the Figure.

Table 1. Specify that you show ranges below averages/means, specify which. Specify what NA means. –

A comment will be added to the end of the Table caption to clarify what these values are "Values given are average $\pm$ standard deviation (range). NA denotes that data is not available."

---

## Author Comment (AC5) · 24 May 2018

A couple of comments follow about aspects of the paper that could be improved. - Each of S. Giovannoni's comments has been addressed individually as follows.

1. I recommend commenting on the abundance of methylophaga and OM43 in the 454 data, or indicate they were not detected if that is the case. It may be that the relatively low coverage obtained in this study (386 seqs/sample) led to these taxa being undetectable. If this is the case, that should be explained so that readers new to this topic understand the issues. OM43 is not mentioned at all, but perhaps it should be, since it has been shown to be an obligate methylotroph, is one of the dominant taxa in some coastal environments, and has been shown to be a source of abundant XoxF

peptides in a coastal ocean metaproteome.

The authors recognise this omission in the manuscript and acknowledge that these taxa should be included to reflect our current understanding of marine methanol utilisation. Therefore, we suggest the addition of the following comments:

"Although numerically very rare (1-11 16S rRNA gene sequences per sample), 16S rRNA gene sequences identified as Methylophaga spp., Methylophaga sp. DMS021 (EU001861) and uncultured Methylophaga sp. (EU031899), were found in each of the Atlantic Ocean provinces in this study (at 97% PAR or 200m depth), consistent with previous identification of Methlophaga spp. in these Atlantic provinces using mxaF gene cloning in (Dixon et al., 2013)." (Page 14, lines 14-19).

"Members of Betaproteobacteria, OM43, have been shown to be obligate methylotrophs, with cultivated cells of strain HTCC2181 dissimilating 3.5 times more methanol than was assimilated (Halsey et al., 2012). OM43 were not successfully identified in the 16S rRNA sequences in this study, which could be an artefact of the relatively low sequence coverage (386 sequences per sample) leading to this taxon not being detectable. During a previous coastal study in the western English Channel (16S rRNA pyrosequence data, Sargeant et al., 2016) only a single sequence of the OM43 clade, HTCC2181, was identified. This is a limitation of this type of environmental sequencing effort and should be a consideration in planning any future projects aiming to understand microbial function through process measurements alongside the generation of metagenomic datasets." (Page 14, line 28 – Page 15, line 6).

This would also require the addition of Halsey et al., 2012 to the full reference list: Halsey KH, Carter AE, Giovannoni SJ (2012) Synergistic metabolism of a broad range of C1 compounds in the marine methylotrophic bacterium HTCC2181. Environmental Microbiology 14:630-640.

2. Amplicon ratios are not as powerful as cell numbers for identifying correlations between taxa and rates, although they are much easier to obtain. So, the correlations

with SAR11 are not with SAR11 cells per unit volume, which would be best, but rather a correlation between the relative success of SAR11 in the community and rates of MeOH oxidation. I suggest the authors revisit the manuscript and choose wording that conveys these issues to oceanographer readers, who often misunderstand this aspect of relative abundance data. –

The authors recognise that this is a limitation and clarity should be provided. We suggest the addition of;

"It should be noted that this correlation has been made with amplicon ratios, relating to the relative success of SAR11 in the community, rather than with SAR11 cell numbers specifically." (Page 13, lines 10-12).

"More work is required to add clarity and understanding to the role that SAR11 cells play in marine community methanol dissimilation." (Page 14, lines 3-4).
* * *

---

## Author Response (AR1)

[revised manuscript text omitted]

**Figure and Table legends.**

**Figure 1.** Remotely sensed MODIS-Aqua chlorophyll *a* composite image of the Atlantic

Ocean from November 2009 (image courtesy of NEODAAS). White squares represent sampling points and ovals indicate samples within different oceanic provinces, labelled with province names NT (northern temperate), NSG (northern subtropical gyre), NTG (northern tropical gyre), EQU (equatorial upwelling), SG (southern gyre), ST (southern temperate).

**Figure 2**. Variability in a) microbial methanol dissimilation rates (using the specific activity of $^{14}CH_3OH$) and b) bacterial leucine incorporation (BLI), in surface waters of the Atlantic

Ocean. Rates were determined from pre-dawn (♦ solid line) and solar noon (◊ dashed line)

CTD casts. Error bars represent ±1 s.d. of triplicate samples, dashed vertical lines indicate

Atlantic province boundaries.

**Figure 3.** Average depth profiles in Atlantic provinces for a) microbial methanol dissimilation (using the specific activity of $^{14}CH_3OH$) and b) bacterial leucine incorporation (BLI) in pre-dawn waters. Error bars represent ±1 s.d. of variation within the province, province averages derived from NT (n = 5), NSG (n = 5), NTG (n = 3), EQU (n = 4), SG (n =

5) and ST (n = 3), except for BLI where there is no data from the ST.

**Figure 4.** Microbial methanol (a) dissimilation and (b) assimilation rates (nmol $l^{-1}$ $h^{-1}$) in the top 200 m of an Atlantic Meridional transect (contour plots).

**Figure 5.** Changes in bacterial community composition (Order, identified using 16S rRNA

gene sequencing) for a) 97 % PAR surface 5m, b) 33 % PAR 10-31m, c) 1 % PAR 15-54m and d) 200 m for different provinces (NT, NSG, NTG, EQU and SG) of the Atlantic Ocean.

Analysis is based on a rarefied sample of 386 sequences per sample. Bacterial Orders individually contributing to less than 5% of the total sample sequences were pooled together into 'Others (<5%)' for clarity. Where ■ *Prochlorococcus*, ■ *Alteromonadales*, □

SAR11 clade, □ *Oceanospirillales*, ■ *Rhodospirillales*, ■ *Flavobacteriales*, □

*Rhodobacterales*, □ *Sphingomonadales*, □ *Synechococcus*, □ *Acidimicrobiales*, □

Order III *Incertae Sedis*, ▨ SAR324 clade (Marine group B), ■ uncultivated bacterium, □

other bacteria individually comprising <5%.

**Figure 6.** Non-metric multi-dimensional scale (MDS) plots of (a) a Bray-Curtis similarity matrix of the 16S rRNA gene sequences of the bacterial community, (b) a Euclidean distance matrix of environmental parameters (salinity, temperature, chl. a, primary productivity, inorganic nutrients, flow cytometry cell numbers, BLI) and (c) a Euclidean distance matrix of rates of methanol dissimilation. Dashed lines highlight significant sample grouping. Plots generated using PRIMER-E (www.primer-e.com). For (a) and (b) ■ represents samples from

200 m i.e. 0 % PAR.

**Table 1.** Summary of rates of methanol uptake (dissimilation and assimilation), methanol
concentrations, bacterial leucine incorporation (BLI) and production (BP), numbers of
heterotrophic bacteria (BN), *Prochlorococcus* (Pros) and *Synechococcus* (Syns).  "Values
given are average ± standard deviation (range).  NA denotes that data is not available."

**Figure 1.**

[Figure]

Chlorophyll *a* (mg m⁻³)

**Figure 2.**

[Figure]

[Figure]

[Figure]

**Figure 3.**

**Figure 4.**

(a)

[Figure]

(b)

[Figure]

**Figure 5.**

[Figure]

**Figure 6.**

(a)

(b)

(c)

[Figure]

**Table 1.** Summary of rates of methanol uptake (dissimilation and assimilation), methanol concentrations, bacterial leucine incorporation (BLI) and production (BP), numbers of heterotrophic bacteria (BN), *Prochlorococcus* (Pros) and *Synechococcus* (Syns). Values given are average ± standard deviation (range).  NA denotes that data is not available.

| | Overall | Atlantic province | | | | | |
|---|---|---|---|---|---|---|---|
| | | NT | NSG | NTG | EQU | SG | ST |
| Methanol dissimilation (nmol L$^{-1}$ h$^{-1}$) | 0.45±0.42 (0.01–1.68) | 0.69±0.35 (0.22–1.50) | 0.99±0.41 (0.15–1.68) | 0.18±0.04 (0.10–0.25) | 0.11±0.03 (0.07–0.17) | 0.24±0.12 (0.01–0.45) | 0.20±0.05 (0.11–0.27) |
| Methanol assimilation (x 10$^{-2}$) (nmol L$^{-1}$ h$^{-1}$) | 0.51±0.54 (0.00–2.24) | 0.54±0.53 (0.00–2.23) | 0.53±0.56 (0.17–1.51) | NA | 0.67±0.66 (0.00–2.24) | 0.19±0.16 (0.00–0.57) | NA |
| BLI (pmol L$^{-1}$ h$^{-1}$) | 9.4±8.9 (0.5–60.2) | 7.7±4.0 (0.9–14.2) | 9.7±14.2 (1.0–60.2) | 8.0±4.3 (2.0–17.0) | 13.7±7.9 (0.6–26.4) | 8.2±9.5 (0.5–41.5) | NA |
| [a]BP(TCF) (ng C L$^{-1}$ h$^{-1}$) | 14.6±13.8 (0.8–96.1) | 11.9±6.1 (1.5–22.0) | 15.0±21.9 (1.5–96.1) | 12.4±6.6 (3.2–26.3) | 21.2±12.2 (1.0–41.0) | 12.7±14.8 (0.8–64.3) | NA |
| [b]BP (ECF) (ng C L$^{-1}$ h$^{-1}$) | 4.8±4.6 (0.3–31.6) | 3.9±2.0 (0.5–7.2) | 4.9±7.2 (0.5–31.6) | 4.1±2.2 (1.0–8.7) | 7.0±4.0 (0.3–13.5) | 4.2±4.9 (0.3–21.1) | NA |
| Numbers of heterotrophic bacteria (x10$^{5}$) (cells mL$^{-1}$) | 6.5±6.3 (1.4–82.6) | NA | NA | 5.8±2.0 (1.6–9.8) | 8.8±10.3 (1.4–82.6) | 5.4±4.4 (1.5–35.8) | NA |
| Numbers of *Prochlorococcus* sp. (x10$^{5}$ cells mL$^{-1}$) | 1.12±4.62 (0.0-4.62) | 0.91±0.07 (0.0-2.56) | 0.89±0.71 (0.0-4.21) | 1.52±1.23 (0.0-4.19) | 1.67±0.2 (0.0-4.62) | 1.20±0.01 (0.0-2.45) | 0.35±0.22 (0.0-2.33) |
| Numbers of *Synechococcus* sp. (x10$^{4}$ cells mL$^{-1}$) | 1.64±31.4 (0.0-31.4) | 1.96±3.61 (0.0-12.7) | 0.18±0.21 (0.0-0.93) | 0.15±0.17 (0.0-0.73) | 1.34±2.69 (0.0-12.8) | 0.14±0.13 (0.0-0.79) | 8.30±10.3 (0.02-31.4) |
| [c]Methanol (nM) | 143±82 (38–420) | 110±126 (38–420) | 203±38 (154–281) | 193±46 (148–278) | 148±37 (117–241) | 110±33 (58–176) | 132 |

[a]Theoretical conversion factor (TCF) 1.55 kg C mol leu$^{-1}$, [b]empirical conversion factor (ECF) 0.51 kg C mol leu$^{-1}$, [c]From Beale et al., (2013)